# Planning of Medical Flexible Needle Motion in Effective Area of Clinical Puncture

**DOI:** 10.3390/s23020671

**Published:** 2023-01-06

**Authors:** Shuai Feng, Shigang Wang, Wanxiong Jiang, Xueshan Gao

**Affiliations:** School of Automation, Guangxi University of Science and Technology, Liuzhou 545006, China

**Keywords:** flexible needle, robotic assistance, motion planning, puncture direction

## Abstract

Lung cancer is the leading cause of cancer deaths worldwide. Although several lung cancer diagnostic methods are available for lung nodule biopsy, there are limitations in terms of accuracy, safety, and invasiveness. Transbronchial needle aspiration (TBNA) is a common method for diagnosing and treating lung cancer that involves a robot-assisted medical flexible needle moving along a curved three-dimensional trajectory, avoiding anatomical barriers to achieve clinically meaningful goals in humans. Inspired by the puncture angle between the needle tip and the vessel in venipuncture, we suggest that different orientations of the medical flexible needle puncture path affect the cost of the puncture trajectory and propose an effective puncture region based on the optimal puncture direction, which is a strategy based on imposing geometric constraints on the search space of the puncture direction, and based on this, we focused on the improved implementation of RCS*. Planning within the TBNA-based lung environment was performed using the rapidly exploring random tree (RRT), resolution-complete search (RCS), and RCS* (a resolution-optimal version of RCS) within an effective puncture region. The experimental results show that the optimal puncture direction corresponding to the lowest cost puncture trajectory is consistent among the three algorithms and RCS* is more efficient for planning. The experiments verified the feasibility and practicality of our proposed minimum puncture angle and puncture effective region and facilitated the study of the puncture direction of flexible needle puncture.

## 1. Introduction

Lung cancer is the leading cause of cancer deaths worldwide, with more people dying each year from lung cancer than prostate, breast, and colon cancers combined [1,2], and has the lowest survival among other common cancers. The fact that most cases of lung cancer are not discovered until late in life is one reason that it is so fatal, and early diagnosis and treatment are effective ways to improve survival. Currently, flexible bronchoscopy or transbronchial needle aspiration (TBNA) is the most common treatment for lung cancer. Different treatment strategies were used to determine the type of lung nodules, and their overall diagnosis and treatment regimen [3] are shown in Figure 1.

Suspected lung nodules in early lung cancer are often exceptionally small and profound inside the lungs, making it difficult for specialists to analyze them precisely. Even though numerous symptomatic strategies are accessible for the biopsy of lung nodules, they have impediments to accuracy, security, or invasiveness.

Transbronchial needle aspiration (TBNA) involves a robot-assisted medical flexible needle. The concept of flexible needle control was first proposed by DiMaio et al. [4,5]. A flexible needle made of a special material with a variable puncture path has become the new favorite in piercing devices. With the minimally invasive, safe, and feasible nature of robotic puncture, robot-assisted flexible needle puncture can skillfully use the flexion of the needle shaft to avoid anatomical obstacles such as nerves, blood vessels, and bones and flexibly and precisely reach target locations that cannot be reached by traditional rigid needles, minimizing surgical trauma and increasing the speed of patient recovery after surgery. At present, research on robot-assisted puncture technology mainly includes three aspects [6]: (1) regional information sensing technology; (2) navigation and interaction control; and (3) puncture and motion planning. This is shown in Figure 2. Among them, the flexible needle puncture motion planning algorithm will be mainly studied here. For flexible needle puncture to be accepted clinically, it is important to verify the correctness and effectiveness of the motion planning algorithm involved in robot-assisted flexible needle automatic puncture from the point of view of patient care, safety, and supervision.

In the obstacle environment, the direction of the starting point will affect the puncture path. Assuming that the needle axis does not rotate, the starting direction of the puncture is the decisive factor. We searched a large number of papers on algorithms related to puncture orientation in the literature and found that scholars stress the importance of puncture orientation. For example, the AHFT (Adaptive Hermite Fractal Tree) proposed by Pinzi et al. [7] was able to determine the region and direction of the puncture site based on the location and direction of the target, but they did not take into account the differences in the optimal solution of the movement path due to the puncture direction when planning the movement. Meng Yu Fu et al. [8] planned the path with RCS* (a resolution-optimal version of resolution-complete search (RCS)) through the coordinates and directions of the given starting and target points. However, when the starting point is different, the growth direction of the tree will be different. Therefore, given the target point and direction, there is no guarantee that the planned path after the current puncture point is optimal. Based on the rapidly exploring random tree (RRT) algorithm, Zhang et al. [9,10] proposed a strategy of adapting some of its parameters for different tissues during insertion, which enhances the security of surgeries. The practicality of their study would be improved if the puncture direction was considered. In summary, existing flexible needle motion planning algorithms do not pay attention to the effect on the puncture path due to differences in the puncture starting direction, and, to the best of our knowledge, such a study is not available in the current literature.

The minimum physiological angle and effective area of puncture are first analyzed and expounded in this paper. A spherical point cloud with a diameter of 1 mm is created at the starting point of the puncture, and the opposite direction of the normal vector of the point cloud is used as the puncture direction. Taking the TBNA (bronchial needle aspiration)-based lung environment as an example, the obstacle area, starting point, and target were established. Finally, the path with the lowest cost was found in the area where the puncture was feasible. The experimental results showed that the best puncture direction was in the area where the puncture direction was effective when the puncture starting point was determined, as shown in Figure 3. In Section 2 below, the research status of relevant algorithms and the principles of puncture kinematics are introduced. The flexible needle puncture is defined and assumed in detail in Section 3. Section 4 describes the improved RCS* algorithm and its enhancements. Section 5 covers simulation studies in a 3D environment to validate the algorithm’s superiority. Section 6 presents our conclusions and the algorithm’s benefits.

## 2. Related Work

### 2.1. Three-Dimensional Static Puncture Motion Planning

The rapidly exploring random tree (RRT) algorithm [11] is a sample-based method for obtaining puncture pathways. Patil et al. proposed an RRT-based needle motion planning algorithm [12] that allows rapid path correction using Lie group symmetry without a clear control strategy, but it samples in 3D workspaces rather than configuration spaces, making the probability of completeness of the original RRT uncertain. To avoid directly addressing curvature constraints in the RRT algorithm, there is a hybrid approach combining sampling and other techniques, such as Favaro et al. [13], who recommend the use of RRT*, a sample-based heuristic search to define the asymptotically optimal solution for path length, followed by a smoothing phase to meet the required kinematic constraints of the needle.

Liu, Fangde, et al. [14] proposed an adaptive fractal tree for needle steering (AFT), using GPU to further accelerate the computation of their method iteratively to extract the cheapest plan from the previous iteration, but the best plan for extracting coarse resolution does not necessarily yield the best plan for higher resolution.

Meng Yu Fu et al. [15] presented the first resolution-complete but not the resolution-optimal manipulable needle motion planner (RCS), for steerable needles. If there is a puncture path, the RCS will find motion planning in a limited time, provided the parameter resolution is good enough, but it does not guarantee motion planning costs. Then, they enhanced the RCS, which explores the state space similarly to A*, with cost-conscious repetitive pruning combined with motion planning costs to improve efficiency. This new approach is called RCS* [8], which is a resolution optimization version of the RCS. It shows excellent practicability in operation time and provides a strong theoretical guarantee for the integrity of motion planning and global optimization in a limited time. Multiresolution motion planning algorithms facilitate future hardware implementations, but the algorithm is essentially static planning, does not consider the relative motion of needle tissue, and only includes the internal biopsy of the bronchoscope.

Hoelscher et al. [16] combined the robustness of inverse planning with the rapidity of the search method to propose a new sampling-based planning method for a steerable needle lung robot, and two speedup strategies were introduced: One strategy is based on imposing geometric constraints on the search space, and the other strategy tree extension toward regions of interest. The effective puncture area proposed in our paper is based on the first strategy. We propose a strategy based on the minimum puncture angle and the effective puncture region, which is based on the imposition of geometrical constraints on the search space in the direction of the puncture. Wu et al. [17] reviewed the development status of flexible steerable needle puncture (FSNP) surgical robots and noted that research on FSNP robots and related technologies is still evolving but lacks depth in many aspects, and so greater efforts and more research are required by those interested in FSNP to promote the early realization of large-scale applications of FSNP technology.

According to the characteristics of the algorithm, the above flexible needle motion planning algorithm is compared in Table 1. Only the RCS* algorithm has the best resolution and is beneficial for hardware implementation. AHFT considers resolutions from coarse to fine but relies on GPU threads. AFT, RRT, and RRT* all have probabilistic completeness due to being sample-based.

### 2.2. Three-Dimensional Dynamic Puncture Motion Planning

The main factor affecting the accuracy of puncture is the target position error caused by needle and soft tissue interaction during puncture. In most minimally invasive surgeries, the margin of error for puncture is generally within millimeters, which can also lead to severe complications [18]. The clinical puncture process is greatly influenced by the unconscious movement of the patient during the procedure (e.g., breathing, heartbeat, muscle pulsation during pain, etc.) and the influence of physiological reactions (e.g., gland swelling), as well as changes in soft tissue properties (e.g., changes in soft tissue material properties at different stages of the lesion) [19]. Therefore, dynamic puncture is more compatible with human physiology than static puncture motion planning. It is a trend to transition static motion planning algorithms to dynamic planning.

Duindam et al. [20,21] first constructed target functions, plotted and optimized needle tail motion trajectories based on distance, tissue damage, and controlled energy parameters, and then used inverse kinematics to solve time-dependent motion planning problems and obtain needle tip optimization pathways without providing theoretical guarantees. Yinoussa et al. [22] presented a new method of automatic steering of robotic needles in deformable tissues, taking into account any deformation of the needle and the trajectory itself. By giving a safe puncture path in advance, according to the deformation of tissue, the points on the needle axis can be obtained by inverse kinematics deconstruction, and the attitude can be adjusted in real time, which is of great significance for robot-assisted puncture control research.

## 3. Definitions and Assumptions

### 3.1. Flexible Needle Puncture Kinematics

Flexible needles made of special materials are controlled by the insertion and rotation of their base to control the trajectory of the needle movement, regardless of the deformation of the tissue during needle penetration. During needle penetration, the needle has only two degrees of freedom, axial displacement *v* and axial rotation ($w_{tip}$ and $w_{base}$), and the asymmetric force exerted by the bevel due to the asymmetry of the piercing needle tip causes the needle to produce a bent trajectory, as shown in Figure 4. The maximum curvature of the needle in the tissue, $k_{max}$, is influenced by the bevel angle of the needle tip and the properties of the tissue [23], and the curvature of the needle motion neglecting the properties of the tissue depends entirely on the needle tip bevel angle [24].

### 3.2. Flexible Needle Puncture Motion Planning

The precise definition of the research problem is helpful to the research. Robot-assisted puncture is a multidisciplinary engineering difficulty, of which motion planning is a branch. Conventional puncture procedures use rigid puncture needles; the flexible needle is a variable curvature puncture needle with the advantage of avoiding anatomical obstacles with the help of a curved trajectory. Therefore, the flexible needle puncture motion planning technology is equivalent to autopilot technology in the automotive field.

We used transbronchial needle aspiration biopsy (TBNA) as an example to construct a lung simulation environment in which vascular, bronchial bone, and other tissues are punctured as obstructions with a known directionally indeterminate starting point and a target. Therefore, the core of flexible needle puncture motion planning is to plan a collision-free path with constant curvature from start to target in a simulated lung environment.

### 3.3. Planning Assumptions for Flexible Needle Puncture Motion

One of the biggest challenges of flexible needle puncture is to ensure that the needle reaches its target location accurately. The movement of tissues and organs in the body, as well as the flexible nature of the needle, make needle movement uncertain and therefore require motion planning to ensure that the needle reaches the target location accurately. Many factors also need to be considered when planning the movement of a flexible needle puncture to ensure that the needle reaches the target location accurately, including the physical properties of the needle, the properties of the soft tissue, the movement of the organ, and the contraction of the muscle. Combining the characteristics of flexible needle puncture, we proposed the following assumptions.

During needle puncture, the shape, size, curvature, and other characteristics of the needle determine the needle movement and directly affect the success rate of needle puncture. The needle is subject to resistance from soft tissues, which leads to changes in the trajectory of the needle movement. Therefore, we assume that the special material of flexible needles used [15,18] is sufficiently flexible. The needle tip bends with maximum curvature $k_{max}$ as it passes through the tissue, and the needle tail trajectory strictly follows the needle tip trajectory, with $w_{tip}$ and $w_{base}$ remaining constant.Soft tissues may have heterogeneous density and strength and may also be subject to external forces, such as respiration or muscle contraction, which may affect needle movement. We assume that the soft tissues have uniform density and strength as well as the same structure and friction coefficient during puncture and that the position of the surrounding tissues and the target site do not move relative to each other during the puncture.During needle puncture, visual sensors or other sensors can be used to measure the needle position and attitude, as well as the needle moment and angular velocity in real-time, and to adjust the needle trajectory to stay on the planned trajectory using the robot’s motion control system. Since the needle trajectory needs to be predicted by planning algorithms, some algorithms with multiresolution properties, such as resolution-complete search (RCS), and RCS* (a resolution-optimal version of RCS), generate paths with very small spaced nodes, which may not be possible by the robot’s motion control system. Therefore, we assume that a control system exists that can adjust the needle trajectory with minimal resolution.

## 4. Methods

### 4.1. Problem Description

Suppose there exists a three-dimensional workspace $W\in R^{3}$ where the obstacle space $W_{obs}\subset W$. Suppose the configuration space $X\subset SE\left(3\right)$ where the pinpoint position is located; then, for any point $x=\left(p,q\right)\in X$, where *p* is the position and *q* is the direction, $p\in R^{3}$, $q\in SO\left(3\right)$. x=(p,q) is collision-free only when $p\notin W_{obs}$, and $X_{free}$ is the set of all collision-free points. Let the trajectory of the needle tip $\sigma :\left[0,l\right]\to$*X*, *l* be the length of σ and denote the length of the trajectory σ trajectory by $l_{\sigma }$ and $\forall s\in \left[0,l\right]$, $\sigma \left(s\right)\in X_{free}$.

Meng Yu Fu et al. [8,25] proposed two methods for calculating configuration space-based costs: One defines the cost of a motion plan as the integral of σ along a given trajectory, i.e., $C\left(\sigma \right)=\int_{0}^{l}c\left(\sigma \left(s\right)\right)ds$. The other is a method to automatically extract a cost map from CT images of the lung, encoding the obstacle, which is associated with the presence of each location in the anatomical structure corresponding to the approximate likelihood of the presence of blood vessels. The first one is used in this paper.

In venipuncture, the success rate and the degree of patient pain are related to the angle of needle entry and the orientation of the needle entry, which is preferable from above the vessel at 40° for better venous conditions and from above or lateral to the vein at 20° for fine and superficial veins [26]. In the same way, during flexible needle TBNA (trans-bronchial needle aspiration), the entry angle of the needle tip into the tissue is not arbitrary, and there exists a certain minimum physiological angle α. When the angle between the entry direction and the surface of the tissue to be punctured is less than α, the puncture is invalid. The yellow area in Figure 5b corresponds to the α region in Figure 5a, and the green area is the effective direction region, expressed in the polar coordinate system as:(1)$$\left\{\begin{array}{c} x_{0}={p_{start}}_{x}+sin\left(\theta \right)cos\left(\varphi \right)\\ y_{0}={p_{start}}_{y}+sin\left(\theta \right)cos\left(\varphi \right)\\ z_{0}={p_{start}}_{z}+cos\left(\theta \right) \end{array}\right.$$
where $\theta \in \left[0,\frac{\pi }{2}-\alpha \right]$, $\varphi \in \left[0,2\pi \right]$, $\left({p_{start}}_{x},{p_{start}}_{y},{p_{start}}_{z}\right)$ is the puncture start point location, and $\left(x_{0},y_{0},z_{0}\right)$ is the effective regional spherical point cloud, which we call $Q_{valid}\subset SO\left(3\right)$. The corresponding normal vectors $\left(\vec{n_{x_{0}}},\vec{n_{y_{0}}},\vec{n_{z_{0}}}\right)$ for each point cloud are:(2)$$\left\{\begin{array}{c} \vec{n_{x_{0}}}=sin\left(\theta \right)cos\left(\varphi \right)\\ \vec{n_{y_{0}}}=sin\left(\theta \right)cos\left(\varphi \right)\\ \vec{n_{z_{0}}}=cos\left(\theta \right) \end{array}\right.$$

The point cloud’s normal vector direction points from the center of the sphere to the surface, like the arrow in red in Figure 5b. The actual puncture direction $\left(\vec{q_{x_{0}}},\vec{q_{y_{0}}},\vec{q_{z_{0}}}\right)$ at the puncture start point in the effective puncture region should be opposite to the point cloud normal vector: (3)$$\left\{\begin{array}{c} \vec{q_{x_{0}}}=-\vec{n_{x_{0}}}\\ \vec{q_{y_{0}}}=-\vec{n_{y_{0}}}\\ \vec{q_{z_{0}}}=-\vec{n_{z_{0}}} \end{array}\right.$$

This is shown by the blue arrow in Figure 5b.

We describe the definition of the piercing movement planning problem in Section 3.2 as follows:

Set static motion planning parameters $\nabla =(X,W_{obs},x_{start},p_{goal},\tau ,\alpha ,l_{\max},k_{\max},C)$, where *X* is the manipulable space of the needle tip, $W_{obs}$ is the obstacle space, $X_{start}$ is the puncture starting point, $p_{goal}$ is the puncture target point, τ is the fault tolerance distance between the needle tip and the target point, τ>0, α is the physiological minimum puncture angle, $l_{max}$ is the maximum puncture depth, $k_{max}$ is the maximum puncture curvature of the flexible needle, and *C* is the trajectory cost.

Set the project $Proj\left(\cdot \right):X\to W$. Using the algorithm, calculate each $Proj\left(x\right)=p$, if $Proj\left(x\right)\notin W_{obs}$; then, the point is collision-free, and all $Proj\left(x\right)\notin W_{obs}$ composed of is a collision-free and curvature-controlled path from the starting point to the target point.

Based on the above definition, the optimal path is defined as $\sigma^{*}=argmin_{\sigma }C\left(\sigma \right)$, provided that σ is valid and the following conditions are satisfied:1.$\sigma \left(0\right)=x_{start}$;2.$l_{\sigma }\leq l_{max}$;3.$k\leq k_{max}$;4.${∥Proj\left(\sigma \left(l_{\sigma }\right)\right)-p_{goal}∥}_{2}\leq \tau$;5.$p\notin W_{obs}$;6.$x_{start_{q}}\in Q_{valid}$.

The mathematical description of the flexible needle motion planning problem improves the logic of the problem. In the following sections, we will show the puncture effect and the corresponding algorithm after embedding the RCS* algorithm into the effective puncture region.

### 4.2. Algorithm Description

RCS* [8] is the first flexible needle puncture motion planning algorithm with resolution-optimal. We upgraded RCS* to Algorithm 1 by using the strategy provided in Section 4.1 for minimum physiological puncture angle and effective puncture area.
**Algorithm 1** Improved RCS***Input:**$W_{obs},x_{start},p_{goal},\tau ,\alpha ,l_{max},k_{max},\delta l_{max},R_{min}$1:θ←{0,π/2,π,3π/2}, $K\leftarrow \left\{0,k_{max}\right\}$2:root $\leftarrow (x_{start},0,0)$3:OPEN $\leftarrow \left\{root\right\}$, CLOSED ←⌀, bestPlan ← NULL4:**for** ValidDirection $(x_{start},p_{goal},\alpha )\;\mathrm{do}$5:        **While not** OPEN.empty() **do**6:              v← OPEN.extract()7:              **if** Valid$\left(v,W_{obs},p_{goal},l_{max}\right)$ **then**8:                   **if** CLOSED.existDuplicate(v) **then**9:                        **if** GoalReached$\left(v,p_{goal},\tau \right)$ **do**10:                             bestPlan.update$\left(v,x_{start}\right)$11:                 **for**
$\;M\in \mathrm{Primitives}(K,\delta l_{max},\theta )$
**do**12:                        OPEN.insert(v⊕M)13:                 CLOSED.insert(v)14:      ifv!=rootthen15:           $\mathrm{for}\;M\in \mathrm{RefinedPrimitives}(M_{v})\;\mathrm{do}$16:                $\mathrm{if}\;\mathrm{ValidResolution}(M,R_{min})\;\mathrm{do}$17:                        OPEN.insert(v.parent⊕M)18:return bestPlan

Algorithm 1 is our improved version of RCS*, which is also resolution-optimal and includes the following ideas:Search Tree: By constructing a search tree Γ=(V,E) with predefined multiresolution motions in the configuration space, the search tree is based on a given start configuration $x_{start}$ and explores the effective motion plan as it unfolds in the configuration space.Reachable Detection (line 7): The curvature restriction removes some nodes that are unlikely to be the final path. It is only necessary that the distance between the target point and the boundary of the reachable region is less than τ, excluding nodes where the needle tip turns more than 90 degrees.Duplicate Detection (line 8): This is the case if there are similar and identical nodes in the search tree to be deleted. v⊕M (line 12) means that the next node will be performed, and *v* will be put into CLOSED (line 13).Motion Primitive: The Motion Primitive is defined as M=(k,δl,δθ), where the curvature is $k\in \left[0,k_{max}\right]$, arc length is δl>0, and needle base is $w_{base}$ rotation angle δθ∈[0,2π).Cutoff Resolution (line 11): The minimum needle insertion $\delta l_{min}$ and the minimum needle axis rotation $\delta \theta_{min}$ are taken as the shear resolution, and the node is not updated when the kinematic element *M* satisfies $R_{min}=\left\{\delta l_{min},\delta \theta_{min}\right\}$.Multithreaded: The task of each thread is to process the nodes extracted from the open list, making it possible to process the nodes in parallel.

Lines 4 and 10 are the improvements made in this paper. For any given puncture start point, if the current puncture direction is valid (line 4), i.e., $x_{start_{q}}\in Q_{valid}$, a sphere of spherical diameter 1 mm is first created with the puncture start point as the center of the sphere and refined to 544 points as possible puncture directions (Figure 6a). The puncture direction shown in Figure 6b is obtained according to the method shown in Equations (1)–(3). The green area in Figure 6b is the valid direction of puncture, corresponding to the green area in Figure 5b. The red color represents the effective puncture direction that may exist after the program is run in the obstacle environment. Each puncture direction corresponds to an optimal path, and the path with the lowest cost is selected at line 10 in comparison with the paths in other directions.

After the puncture direction is feasible, nodes are iteratively extracted from the OPEN list (line 5). Given an extracted node *v*, we first check that it is valid (line 6) using the condition described by RCS* to ensure that the insertion length does not exceed $l_{max}$ and that the target region is reachable after node *v* differs from another node, where the nodes are sorted according to their rank and secondary metric. In addition, RCS* checks if the cost is less than the cost of the best plan found thus far to reach the target region (line 10) and updates the best paths for all puncture directions thus far.

## 5. Results

We use lung environment point clouds to validate the improvements proposed in this paper for RCS*. All experiments were performed on an Intel(R) Core(TM) i5-7200U CPU (2.50 GHz × 4) with 8G RAM and Ubuntu 18.04.6. Puncture start point coordinates: $x_{start}\left[\left(64.83240,139.4900,-654.0964\right),\left(q_{start_{x}},q_{start_{y}},q_{start_{z}}\right)\right]$, target point $x_{goal\_1}\left[\left(-105.8713,158.2018-613.2506\right),\left(0.9141270,-0.1681980,-0.3688930\right)\right]$ and $x_{goal\_2}\left[\left(-115.8713,148.2018,-603.2506\right),\left(0.4141270,-0.3681980,-0.1688930\right)\right]$. The flexible needle: $k_{max}$ = 50 mm${}^{-1}$, $l_{max}$ = 100 mm, diameter *d* = 2 mm, α = 20°, and set the target tolerance, τ = 1.0 mm. All runs resulted in the smallest puncture trajectory length within 1 s of the run.

Our strategy of imposing punctured effective regions on the search space was applied to a rapidly exploring random tree (RRT), resolution-complete search (RCS), and RCS* (a resolution-optimal version of RCS). Figure 7 shows the effect of the planning of two different targets for the improved RCS* proposed in this paper, with dark blue representing the best puncture path for $x_{goal\_1}$ and red representing the best puncture path for $x_{goal\_2}$. Figure 8, Table 2 and Table 3 show the effective puncture directions and their detailed results. The blue dots in Figure 8 indicate the directions of the existence of valid paths obtained by different algorithms calculated in the current lung environment, reflecting the capabilities of different algorithms. Figure 8a–c list the effective puncture regions (green) corresponding to $x_{start}$ and $x_{goal\_1}$, and the blue dots correspond to the directions of different algorithms in Table 2. Similarly, (d), (e) and (f) represent the case when the starting point position is $p_{start}$ and target point is $x_{goal\_2}$.

Table 2 lists the costs of the planning results of the three improved algorithms from the starting point $x_{start}$ to the target points $x_{goal\_1}$ and $x_{goal\_2}$, where the planning cost represents the puncture trajectory length, and $\left(q_{start_{x}},q_{start_{y}},q_{start_{z}}\right)$ is the value of $\left(\vec{q_{x_{0}}},\vec{q_{y_{0}}},\vec{q_{z_{0}}}\right)$. Taking RCS* in Table 2 as an example, there are 10 directions of the planned path corresponding to 9 blue points and 1 red point in Figure 8b, and this red point also corresponds to the bolded part of the 6th row of RCS* in Table 2. According to Equations (1)–(3), we can also obtain the coordinates of the sphere of this red point in Figure 8b:



$\begin{array}{cc} {\left(x_{0},y_{0},z_{0}\right)}_{red} & =\left(q_{start_{x}},q_{start_{y}},q_{start_{z}}\right)+\left(p_{start_{x}},p_{start_{y}},p_{start_{z}}\right)\\ & =\left(-64.83240,139.4900,-654.0964\right)+\left(0.8923680,-0.2382610,0.3832890\right)\\ & =\left(-63.94000,139.2510,-654.4790\right) \end{array}$



In addition, the red point in Figure 8b is the best puncture direction, which corresponds to the planned trajectory of the blue puncture path in Figure 7, and the starting point and target point of the trajectory are blue and pink, respectively. The other nine blue points correspond to the purple puncture path in Figure 7. Similarly, the same applied to RCS and RRT in Table 3. The difference between Table 2 and Table 3 is the change in the target location for $x_{goal\_2}$. It is found that the rapidly exploring random tree (RRT), resolution-complete search (RCS), and RCS* (a resolution-optimal version of RCS) algorithms run with the direction of (0.892368, −0.238261, −0.383289) (bolded in Table 2) have the lowest cost of puncture (puncture trajectory length), with the RCS* algorithm giving the best results. Figure 8d–f list the effective puncture regions (yellow) corresponding to $x_{start}$ and $x_{goal\_2}$, and the blue points correspond to the directions of the different algorithms in Table 3. It is found that the RCS, RCS* and RRT algorithms run with the direction of (0.924466, 0.001275, −0.381263) (bolded in Table 3) have the lowest cost of puncture (puncture trajectory length), with the RCS* algorithm giving the best results.

The six sets of “Cost” in Table 2 and Table 3 have been integrated into two line charts (Figure 9a,b, respectively). If we do not dilute the spherical point cloud to 544, it will be a set of concave curve graphs. Our proposed effective puncture area was similar to the range of gray dotted lines in Figure 9a,b. Our approach is to move the dotted area to the best evaluation interval or reduce the *d* value by geometrical constraints.

Most of the RCS* in Figure 9a,b are below the RCS or RCS* curves, due to the different sample sizes of each algorithm and the 1 s per planning time frame, which we abandoned to estimate the total cost by integrating the curves with the x-axis. Critically, Meng Yu Fu’s experiments [15] have shown that RCS* exhibits significant performance in terms of actual total consumption cost. The average and cost before reaching the lowest of the six groups in Table 2 and Table 3 have been integrated into Table 4. Based on the characteristics of the data in Table 4 and the trends in Figure 9a,b, we decided to consider the average cost to describe the planning efficiency of the algorithm and the cost before reaching the minimum cost to describe the planning rapidity of the algorithm.

The experiments in this paper revolve around the point cloud, and the lung environment is used as an example to establish the obstacle area, the starting point and the target point. A spherical point cloud with a spherical diameter of 1 is established with the puncture starting point as the center of the sphere, and the inverse of the point cloud normal vector is used as the puncture direction. We focus on the RCS* described, using the rapidly exploring random tree (RRT), resolution-complete search (RCS), and RCS* (a resolution-optimal version of RCS) algorithms in the puncture feasible region, which is the effective region proposed in this paper. In the effective region, the experimental data of puncture planning for all effective puncture directions of two different target points and starting points are analyzed by three algorithms to find the path with the smallest puncture trajectory from the starting point to the target point, whose corresponding direction is the best puncture direction, and to explore the connection between them and find the law. The experimental results found that the optimal puncture trajectory corresponding to the puncture direction is the same for the three algorithms.

Figure 7 shows the planning effect of the puncture effective region proposed in this paper to improve RCS*, and the method of the effective region proposed in this paper can also be transferred to RCS and RRT. Table 2 and Table 3 show the planning effect of these three algorithms through data, and the results show that the best puncture direction of each algorithm planning result is consistent, which directly reflects the rationality and the best puncture direction of the puncture effective region and the best puncture direction proposed in this paper. Figure 8 and experimental data show that the three algorithms have the best RCS* and the second-best RRT in terms of trajectory cost. In summary, the optimal (trajectory length) puncture trajectory exists in the effective region corresponding to the puncture direction $\left(q_{start_{x}},q_{start_{y}},q_{start_{z}}\right)$ in the case where the puncture start point $p_{start}$ is determined.

## 6. Conclusions

Inspired by the puncture angle between the needle tip and the vessel in venipuncture, in this paper, we suggest that the different orientations of the medical flexible needle puncture path will affect the cost of the puncture trajectory and conducted an experimental study. We proposed the optimal puncture direction, in which geometric constraints are imposed to propose an effective puncture region.

The improved RCS* algorithm was tested in a simulated lung environment, and the comparison of the planning data of rapidly exploring random tree (RRT), resolution-complete search (RCS), and RCS* (a resolution-optimal version of RCS) algorithms showed that the optimal puncture direction was found to exist within the valid region of the puncture direction that determined the puncture starting point, and the optimal puncture direction was consistent among the three algorithms. The higher planning efficiency of the RCS* was found by comparing the average cost and the cost before achieving the optimal direction. The experiments verified the feasibility and practicality of our proposed minimum puncture angle and puncture effective region and facilitated the study of the puncture direction of flexible needle puncture. We have built a set of devices that can achieve fine resolution, and in the future, we will subject RCS* to dynamic puncture motion planning.

## Figures and Tables

**Figure 1 sensors-23-00671-f001:**
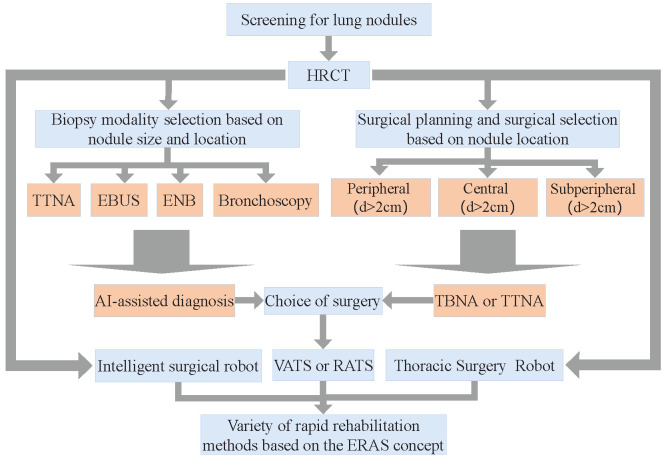
General surgical planning based on pulmonary nodules. Techniques such as High-Resolution CT (HRCT), endobronchial ultrasonography (EBUS), and electromagnetic navigation bronchoscope (ENB) are commonly used for lung cancer screening. Transthoracic needle aspiration (TTNA) and bronchial needle aspiration (TBNA) are commonly used for lung cancer treatment; the former is less invasive but requires the needle to be passed through the chest wall from outside the body, which may lead to pneumothorax; the latter is less dangerous but often fails to reach the outer regions of the lung.

**Figure 2 sensors-23-00671-f002:**
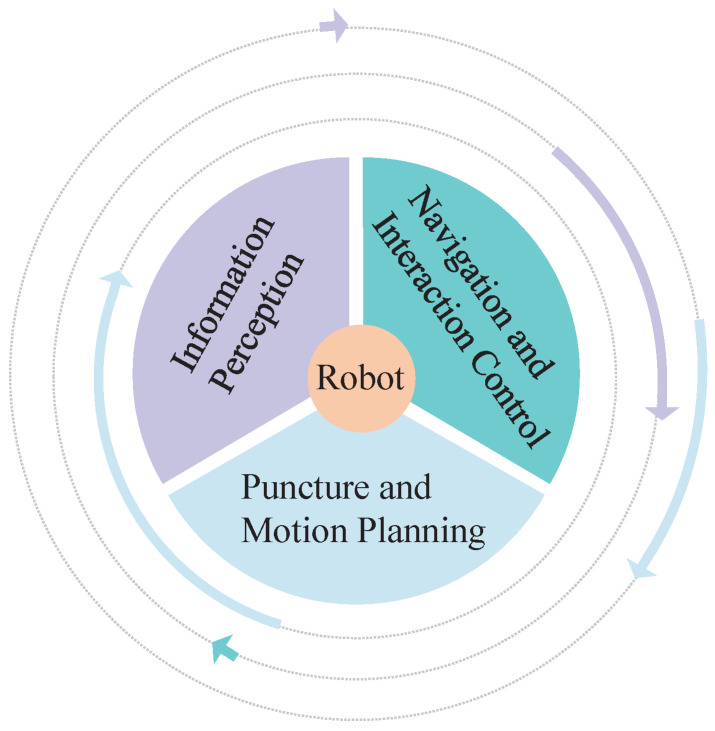
Three key technologies for robot-assisted puncture.

**Figure 3 sensors-23-00671-f003:**
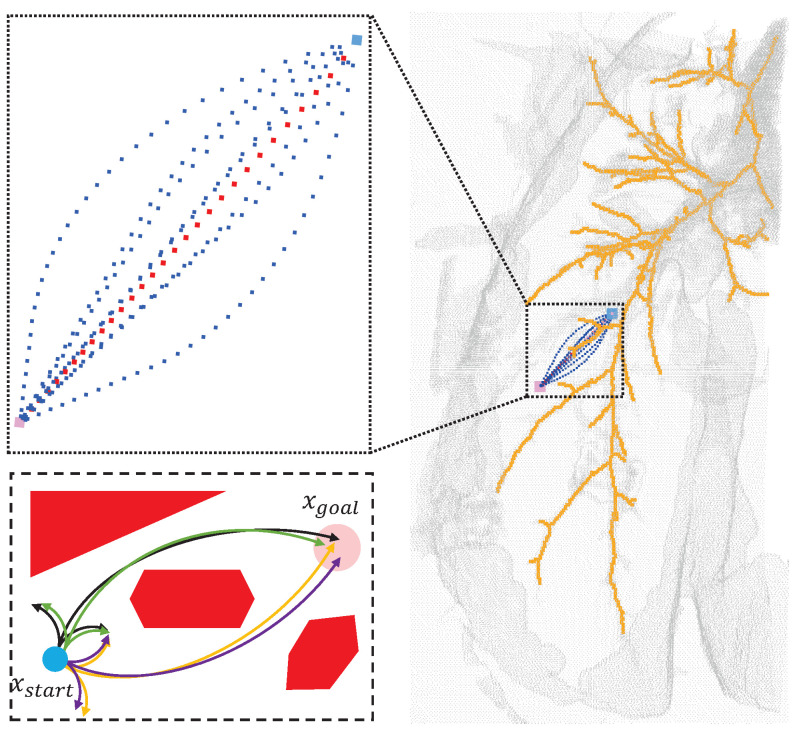
(**Right**): Effect of different puncture directions at fixed puncture points (light blue), both reaching the nodes of the lung parenchyma (pink) while avoiding important anatomical structures such as bronchi (yellow) and other tissues (gray). (**Top left**): Enlarged view of the effect after RCS* planning for different puncture directions. Each trajectory corresponds to the best trajectory for a certain puncture direction, with the red trajectory corresponding to the best direction. (**Bottom left**): The paths after planning for different puncture directions (black, green, yellow, purple) at the starting point (blue) can all reach the target point (pink) while avoiding obstacles (red), and the path corresponding to the lowest cost is chosen as the best puncture direction.

**Figure 4 sensors-23-00671-f004:**
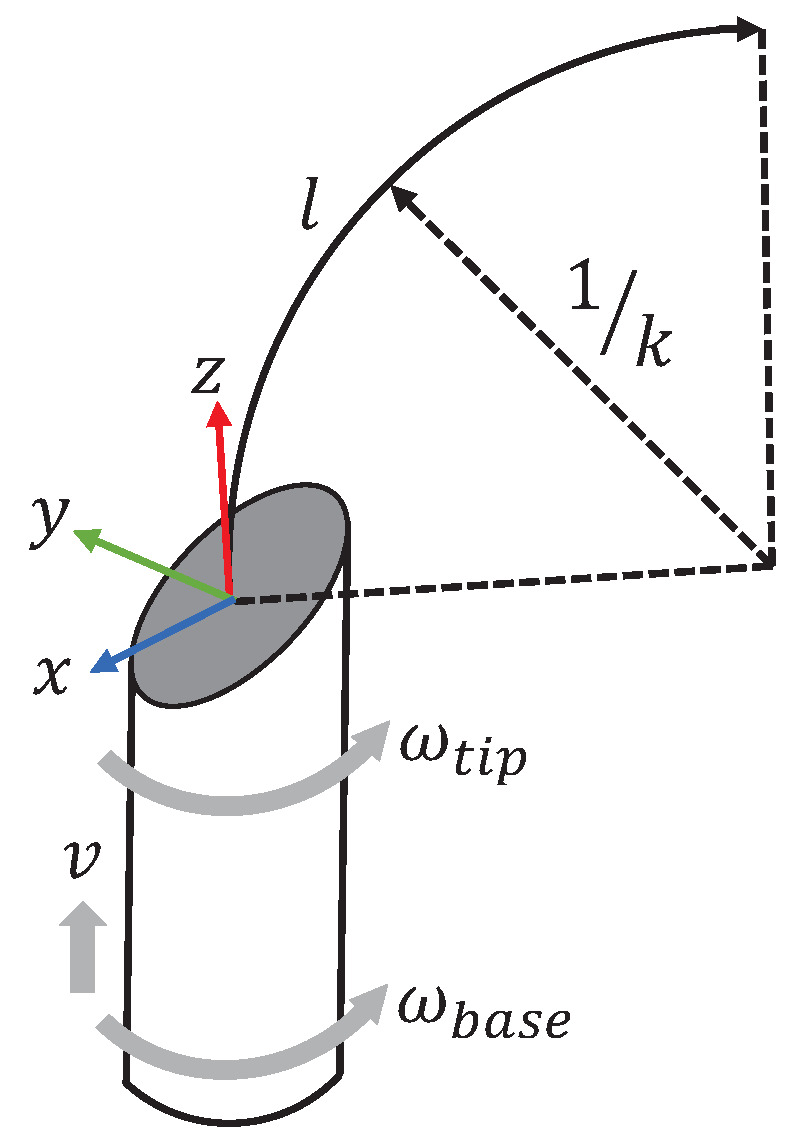
The kinematics of a bevel-tip flexible needle.

**Figure 5 sensors-23-00671-f005:**
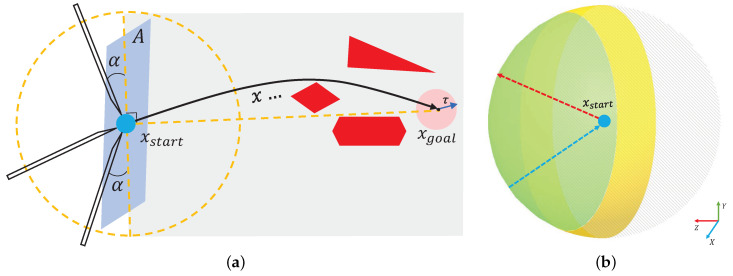
Feasible angle and effective area for flexible needle puncture. (**a**) describes the minimum puncture physiological angle α. A denotes the punctured plane, gray represents the tissue, red represents the obstacle and τ is the target tolerance. (**b**) describes the effective puncture region (green), the yellow region corresponds to the α region in (**a**), and the point cloud normal vector (red) points from the center of the sphere to the sphere with the opposite puncture direction (blue).

**Figure 6 sensors-23-00671-f006:**
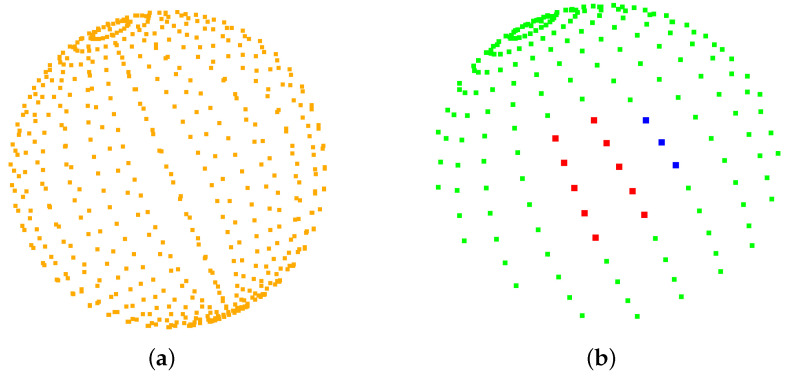
All possible puncture directions are shown in (**a**). In (**b**), all valid directions are shown in green, with red in the obstacle environment denoting the direction in which a viable puncture path is present and blue denoting the direction in which a viable puncture path is absent.

**Figure 7 sensors-23-00671-f007:**
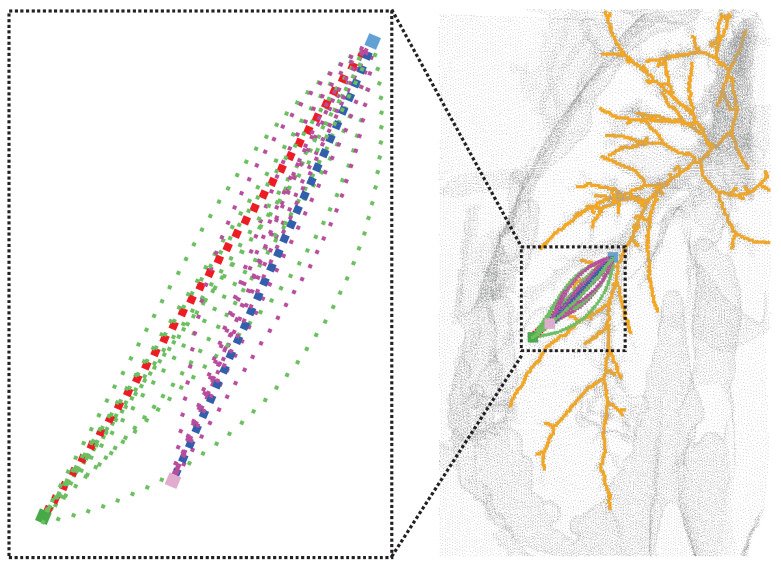
Results after RCS* planning, where the starting point (light blue) and all light green represent all valid paths of $x_{goal\_2}$ (dark green). Purple represents all valid puncture paths of $x_{goal\_1}$ (pink). Red and dark blue are the best paths.

**Figure 8 sensors-23-00671-f008:**
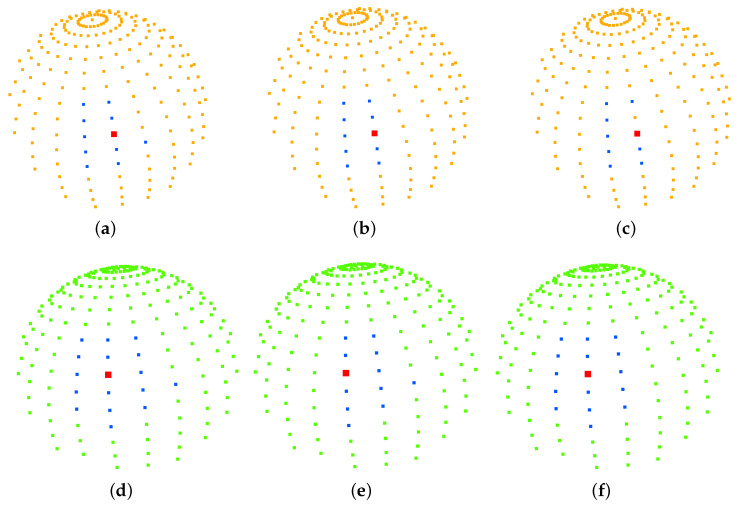
(**a**–**c**) represent the directions of the effective paths (blue) and the best puncture directions (red) obtained by running the algorithm with the starting point position $p_{start}$ and the puncture effective direction (yellow) at the target point $x_{goal\_1}$ for RCS, RCS* and RRT. Similarly, (**d**–**f**) represent the case when the starting point position is $p_{start}$ and target point is $x_{goal\_2}$.

**Figure 9 sensors-23-00671-f009:**
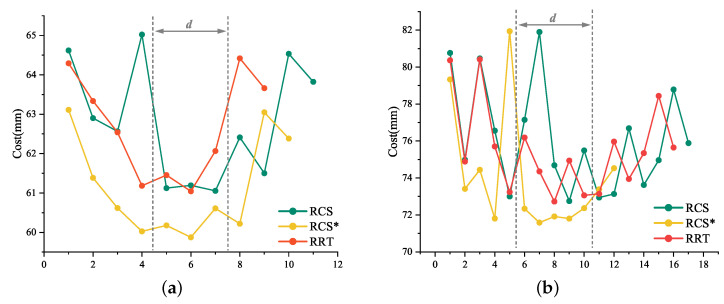
Puncture trajectory cost line chart. (**a**) represents the line graph of three sets of “Cost” in Table 2 and similarly, (**b**) represents the line chart of “Cost” in Table 3.

**Table 1 sensors-23-00671-t001:** Comparison table of characteristics of flexible needle motion planning algorithms.

Characteristic	AFT	AHFT	RRT	RRT*	RCS	RCS*
Completeness	✕	✓	✕	✕	✓	✓
Probabilistic completeness	✓	✕	✓	✓	✕	✕
Optimality	✓	✕	✕	✕	✕	✓
Asymptotic optimality	✓	✓	✕	✓	✕	✕
Optimal resolution	✕	✕	✕	✕	✕	✓

In the column of application, “✓” means yes, “✕” means no.

**Table 2 sensors-23-00671-t002:** Costs of planning results for different puncture directions at starting point position $p_{start}$ and target point $x_{goal\_1}$.

	CS				RCS*				RRT			
	Direction $\left(q_{{\mathrm{start}}_{x}},q_{{\mathrm{start}}_{y}},q_{{\mathrm{start}}_{z}}\right)$			Cost	Direction $\left(q_{{\mathrm{start}}_{x}},q_{{\mathrm{start}}_{y}},q_{{\mathrm{start}}_{z}}\right)$			Cost	Direction $\left(q_{{\mathrm{start}}_{x}},q_{{\mathrm{start}}_{y}},q_{{\mathrm{start}}_{z}}\right)$			Cost
1	0.991465	0.001096	−0.130369	64.6193	0.991465	0.001096	−0.130369	63.1118	0.991465	0.001096	−0.130369	64.2937
2	0.957829	−0.256209	−0.130076	62.9014	0.957829	−0.256209	−0.130076	61.3853	0.957829	−0.256209	−0.130076	63.3354
3	0.965696	0.001197	−0.259673	62.5639	0.965696	0.001197	−0.259673	60.6194	0.965696	0.001197	−0.259673	62.5386
4	0.837163	−0.482563	−0.257470	65.0225	0.933537	−0.249225	−0.257673	60.0265	0.933537	−0.249225	−0.257673	61.1850
5	0.933537	−0.249225	−0.257673	61.1241	0.924466	0.001275	−0.381263	60.1770	0.924466	0.001275	−0.381263	61.4550
6	0.924466	0.001275	−0.381263	61.1949	**0.892368**	**−0.238261**	**−0.383289**	**59.8760**	**0.892368**	**−0.238261**	**−0.383289**	**61.0429**
7	**0.892368**	**−0.238261**	**−0.383289**	**61.0548**	0.865851	0.001334	−0.500299	60.6104	0.865851	0.001334	−0.500299	62.0652
8	0.865851	0.001334	−0.500299	62.4123	0.836002	−0.223848	−0.500992	60.2192	0.792878	0.001369	−0.609379	64.4184
9	0.836002	−0.223848	−0.500992	61.5018	0.792878	0.001369	−0.609379	63.0487	0.766228	−0.205047	−0.608974	63.6609
10	0.792878	0.001369	−0.609379	64.5335	0.766228	−0.205047	−0.608974	62.3829				
11	0.766228	−0.205047	−0.608974	62.3829								

**Table 3 sensors-23-00671-t003:** Costs of planning results for different puncture directions at starting point position $p_{start}$ and target point $x_{goal\_2}$.

	RCS				RCS*				RRT			
	Direction $\left(q_{{\mathrm{start}}_{x}},q_{{\mathrm{start}}_{y}},q_{{\mathrm{start}}_{z}}\right)$			Cost	Direction $\left(q_{{\mathrm{start}}_{x}},q_{{\mathrm{start}}_{y}},q_{{\mathrm{start}}_{z}}\right)$			Cost	Direction $\left(q_{{\mathrm{start}}_{x}},q_{{\mathrm{start}}_{y}},q_{{\mathrm{start}}_{z}}\right)$			Cost
1	0.999999	0.000978	−0.000978	80.7644	0.999999	0.000978	−0.000978	79.3278	0.999999	0.000978	−0.000978	80.3665
2	0.991465	0.001096	−0.130369	74.9835	0.991465	0.001096	−0.130369	73.4151	0.991465	0.001096	−0.130369	74.8899
3	0.957829	0.256209	−0.130076	80.4673	0.957829	−0.256209	−0.130076	74.4401	0.957829	0.256209	−0.130076	80.4152
4	0.957829	−0.256209	−0.130076	76.5572	0.965696	0.001197	−0.259673	71.8050	0.957829	−0.256209	−0.130076	75.7056
5	0.965696	0.001197	−0.259673	73.0024	0.837163	−0.482563	−0.257470	81.9360	0.965696	0.001197	−0.259673	73.2365
6	0.933537	0.249225	−0.257673	77.1477	0.933537	−0.249225	−0.257673	72.3398	0.933537	0.249225	−0.257673	76.1870
7	0.837163	−0.482563	−0.257470	81.8952	**0.924466**	**0.001275**	**−0.381263**	**71.5848**	0.933537	−0.249225	−0.257673	74.3561
8	0.933537	−0.249225	−0.257673	74.6854	0.892368	−0.238261	−0.383289	71.9168	**0.924466**	**0.001275**	**−0.381263**	**72.7260**
9	**0.924466**	**0.001275**	**−0.381263**	**72.7510**	0.865851	0.001334	−0.500299	71.8018	0.892368	0.238261	−0.383289	74.9402
10	0.892368	0.238261	−0.383289	75.4871	0.836002	−0.223848	−0.500992	72.3705	0.892368	−0.238261	−0.383289	73.0592
11	0.892368	−0.238261	−0.383289	72.9416	0.792878	0.001369	−0.609379	73.3859	0.865851	0.001334	−0.500299	73.1431
12	0.865851	0.001334	−0.500299	73.1358	0.766228	−0.205047	−0.608974	74.5285	0.836002	0.223848	−0.500992	75.9681
13	0.836002	0.223848	−0.500992	76.6867					0.836002	−0.223848	−0.500992	73.9458
14	0.836002	−0.223848	−0.500992	73.6244					0.792878	0.001369	−0.609379	75.3445
15	0.792878	0.001369	−0.609379	74.9628					0.766228	0.205047	−0.608974	78.4385
16	0.766228	0.205047	−0.608974	78.7849					0.766228	−0.205047	−0.608974	75.6471
17	0.766228	−0.205047	−0.608974	75.8855								

**Table 4 sensors-23-00671-t004:** Cost statistics for puncture trajectories.

	Table 2-Cost				Table 3-Cost		
	RCS	RCS*	RRT		RCS	RCS*	RRT
Mean	62.79560	61.14570	62.66610		76.10370	74.07100	75.52310
Before Lowest	438.4809	365.1960	373.8506		692.2541	524.8486	607.8828

## Data Availability

Not applicable.

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
