# Peer review of "Planning of Medical Flexible Needle Motion in Effective Area of Clinical Puncture"

_sensors, 2023, doi:10.3390/s23020671_

Round 1

Reviewer 1 Report

The authors present an interesting work in the field of path planning of medical flexible needle motion.

The related work referenced on the paper includes relevant works. The authors could consider the following to complement the scope of the state of the art.

Fu, M., Solovey, K., Salzman, O., & Alterovitz, R. (2022, May). Resolution-optimal motion planning for steerable needles. In 2022 International Conference on Robotics and Automation (ICRA) (pp. 9652-9659). IEEE.

Jiang, S., Jiang, B., Fang, P., & Yang, Z. (2022). Pre-Operative Motion Planner for Steerable Needles Using Cost Map Based on Repulsive Field and Empirical Model of Needle Deflection. Journal of Medical Devices, 16(2), 021004.

Wu, K., Li, B., Zhang, Y., & Dai, X. (2022). Review of research on path planning and control methods of flexible steerable needle puncture robot. Computer Assisted Surgery, 27(1), 91-112.

Zhang, Y., Qi, Z., & Zhang, H. (2021, November). An improved RRT* algorithm combining motion constraint and artificial potential field for robot-assisted flexible needle insertion in 3D environment. In 2021 3rd International Conference on Industrial Artificial Intelligence (IAI) (pp. 1-6). IEEE.

Zhang, Y., Ju, Z., Zhang, H., & Qi, Z. (2021). 3-D Path Planning Using Improved RRT* Algorithm for Robot-Assisted Flexible Needle Insertion in Multilayer Tissues. IEEE Canadian Journal of Electrical and Computer Engineering, 45(1), 50-62.

Hoelscher, J., Fu, M., Fried, I., Emerson, M., Ertop, T. E., Rox, M., ... & Alterovitz, R. (2021). Backward planning for a multi-stage steerable needle lung robot. IEEE Robotics and Automation Letters, 6(2), 3987-3994.

Section 2.3, which refers to kinematics, fits better in section 3 rather than section 2.

The authors should provide the names of the algorithms before referring to them only by their acronyms. Example, Rapidly-exploring Random Tree (RRT).

The ending of section 1, lines 62 and 63 should be improved in the writing.

Reference 13 needs to be updated as the manuscript was published in the proceedings of the Robotics: Science and System XVII virtual conference ISBN 978-0-9923747-7-8

The summarised results in tables 2 and 3 are hard to relate to the starting point and the desired goal.

In the case of  Table 2 the first four values of qstartz for the RCS* and RRT use only five decimal places instead of 6 like the rest of the table.

Furthermore, considering that at 1 all the algorithms start their search as it is the common starting point, the reader has to assume that the last set of direction coordinates listed for each algorithm corresponds to the desired goal. In Table 2  only the RCS* and the RRT arrive to the same goal while the RCS seems to have arrived somewhere else at coordinate 11.

Line 259 makes reference to “bolded” text in Table 1 when it should be table 2. Same legacy reference in line 264.

Although for both examples, it is highlighted in section 5, when discussing the results, that the three algorithms at the particular bolded point the three algorithms traverse through the same direction that coincides of having the lowest cost regardless of the algorithm, it is noticeable that there are other common directions between the three algorithms. Perhaps the authors should consider addressing the total cost of the planned trajectory if relevant to the case, considering that in the two cases, the RCS* provides the shorter paths according to the number of directions listed in tables 2 and 3.

Section 6 should be improved with more solid arguments to sustain the claims made by the authors. For example: “… it was found that the existing flexible needle motion planning algorithm did not pay attention to the study of the direction of the puncture starting point…”, as a reader, it is difficult to agree or disagree with this claim as there is no discussion that precedes it.

Reviewer 2 Report

A very limited literature review is added. I think authors should refer to more latest research. 

Some insight into experimentation may also be provided. 

overall English of the manuscript needs revision. 

Reviewer 3 Report

The paper presents an improved RCS*algorithm used to determinate the best puncture direction of a flexible needle in clinical puncture.

The paper has been examined for multiple times in order to make a valid review and the following had been constated:

1. The last part of the abstract should be reworded because the aim of the papers is confusing and doesn’t offer a relevant view of the work. 

2. The structure of the paper should be improved considering:

-The related work section which includes the description of other approaches used to calculate the trajectory, a comparation of the algorithms used with the same aim and a description of the flexible needle puncture kinematics. There are to many different notions covered in this section. 

- Definitions and assumption section- This section presents to many general details which could be hardly introduce in introduction section perhaps, but definitely these details don’t deserve their own section. 

3. The personal contribution of authors it’s difficult to evaluate. I suggest an improve description of the contribution which can also be certified by the results. 

Other comments are related to 

- the English level considering that the paper has multiple propositions without a verb. The expressions are very confusing, and this makes the reading process very difficult.

- the presentation of images (for example in Figure 6, Figure (b) is bigger than image (a) but because (b) represents the valid directions, and (a) represents the possible directions, I think that these images should have the same size because are related, and in the present form don’t offer relevant information. 

I would also recommend more attention to acronyms (because are multiple acronyms without an explanation), references (for reference no 12, in text, the authors referred to an author who doesn’t correspond to the list of authors present at position 12 in references list), spelling mistakes. 

Round 2

Reviewer 3 Report

The authors have addressed the major concerns expressed in the first set of comments in a very detailed and comprehensive manner, improving the paper towards a completely new version which, in my opinion, can be now considered for publication. 

As a result, my new recommendation is Accept.